# Why Are Corporations Willing to Take on Public CSR? An Organizational Traits Approach

**Yun Liu [1], Greg. G. Wang [2,3] and Yu Chen [1,\*]**

[1]  School of Business Administration, Shanghai Lixin University of Accounting and Finance, Shanghai 201620, China; liuyundevy@sina.com

[2]  Department of Human Resource Development, Soules College of Business, The University of Texas at Tyler, Tyler, TX 75799, USA; GWang@uttyler.edu

[3]  School of Management, Northwestern Polytechnical University, Xi'an 710129, China

[\*]  Correspondence: chenyu@lixin.edu.cn; Tel.: +86-138-1797-9240

**Abstract:** Corporation social responsibility includes the relational responsibility for the contractual stakeholders (relational CSR) and the public responsibility for the whole society (public CSR). In this paper, we examined the effect of organizational virtuousness on a corporation's public CSR behavior and the moderating effect of organizational identity orientation between them. To test our hypothesis, we collected and analyzed a sample from 88 corporations and 742 respondents through questionnaires. Our results show that organizational virtuousness is positively associated with a corporation's public CSR behavior, and this positive effect is moderated by organizational identity orientation. Among them, individualistic and collectivistic identity orientation positively moderates the relationship between organizational virtuousness and public CSR, while relational identity orientation negatively moderates the relationship between them. Our results suggest that a virtuous corporation does not necessarily have more willingness to take on public CSR than its counterparts, because the intention also depends on the type of identity orientation possessed by the virtuous corporation. In order to improve the enthusiasm of enterprises to take on public CSR, in addition to cultivating the virtue of organizations, different management measures should be taken according to the identity orientation of organizations.

**Keywords:** organizational virtuousness; organizational identity orientation; public CSR

## 1. Introduction

Previous research noted that corporate identity consists of normative and utilitarian components with different foci [1]. As a profit-seeking entity, the corporation is a system of stakeholder groups with complex financial relationships between interest groups with respective rights, objectives, expectations, and responsibilities [2]. As such, the corporation's survival and continuing development mostly depend on its ability to take corresponding responsibilities for those stakeholder groups [2]. Those stakeholders who have financial links with the corporation are also known as contractual stakeholders, immediate or direct stakeholders, such as employees, customers, suppliers, shareholders, and so on [3]. At the same time, the corporation is a social entity and citizen of its residing community, taking advantages of local resources for its operations and being expected to make contribution for sustainable development of the local community [4]. Along with the dual corporate identity, the concept of CSR was conceptualized in two dimensions. One is relational CSR, representing the responsibility that it must take for various immediate, direct, or contractual stakeholder groups, and the other is public CSR, signifying its responsibility for the society or community [5]. The relational CSR is more or less

transactional, instrumental, and compulsory, whereas the public CSR is in the ethical, philanthropic, and discretional domain [5].

This article is only focused on public CSR. The first reason for this is because our study is in line with the recent shift in the CSR literature, from examining CSR as an aggregated phenomenon to focusing on a particular dimension [6]. Secondly, Public CSR is important for social development, corporations might trigger social enhancement by engaging in public CSR initiatives, these initiatives include actions within the firm, such as changing methods of production to reduce environmental impacts or changing labor relationships both within the firm and across the firm's value chain, as well as actions outside the firm, such as making infrastructure investments in local communities or developing philanthropic community initiatives [7]. So, more and more researchers today are getting interested in how corporations generate long-term profitability and contribute to the common good to the society [8].

We start our inquiry with the following question: why would a corporation be willing to take public CSR? In reviewing the CSR literature, instrumental, normative, and moral perspectives have been used to explain why for-profit organizations will make voluntary contributions to serve public purposes. Based on resource dependence theory [9], instrumental perspective believes that consistent public CSR activities can promote a corporation's reputation and image that in turn can be translated into the firm's competitive advantage [10] and moral capital [11]. Moral capital, acting as a societal license, becomes a form of insurance which can prevent unforeseen risks in corporate image, reputation, and eventually, profits. The normative perspective was based on neo-institutional theory [12]. It suggests that corporations in a given institutional environment have relational motivation to engage in CSR practices in order to be seen as legitimate through complying with the specific norms, values, and beliefs in the societal environment [13]. Finally, the moral perspective was based on the social contract theory, emphasizing the principle of morality and justice [14]. This perspective believes that corporations has been endowed a moral identity by the society, and this moral identity requires a corporation as a citizen should take an ethical responsibility for the society, no matter how it has been fulfilled and how it should be fulfilled [15,16].

While, in our opinion, activities such as public CSR behavior can be seen as resulting not only from external demands, but also from the corporations' internal personality, namely the activities are the external expression of corporations' internal personality trait. In this study, we intend to reveal the motivation of corporations to take public CSR from the perspective of organizational traits with evidence from China. The remainder of the paper is organized as follows. First, the theoretical background and research hypotheses are presented and proposed. Next, the research methodology is presented, followed by the empirical analyses and results. Subsequently, managerial implications are discussed. Finally, main conclusions are drawn and the limitations of this study and suggestions for future research are discussed.

## 2. Theoretical Framework and Hypotheses

### 2.1. Theoretical Framework

Regarding the issue of what kind of corporations with what traits are more willing to take public CSR regardless of the returns, researchers in positive organizational scholarship attempted to offer explanations based on virtue ethic theory in the past decade. For example, Arjoon [17] argued that public CSR could be equated to the practice of the virtue of "mercy" and viewed public CSR as corporate mercy aiming at promoting the common good. Bert van de Ven [18] also explained corporations' public CSR behavior from an organizational virtue point of view. Bright [19] further notes that, because organizational virtuousness is characterized by unconditional social betterment that extends beyond merely self-interested benefit and creates social value transcending the instrumental desires of the actor, so organizational virtuousness contributes to the genuineness for corporations' public CSR activities.

Yet, existing research only proposed that organizational virtuousness was a critical precondition for corporations to exhibit genuine responsible behaviors; few had empirically examined the

hypothesized relationship between organizational virtuousness and public CSR behavior. Genuineness is a state describing that an organization's CSR activities are well-intentioned and not merely for its instrumental gains [11]. Yet, "well-intentioned" is different from "highly-motivated", thus high genuineness in public CSR may not necessarily mean high motivation in public CSR. So, whether or not a virtuous company is willing to take more public CSR than others, we think it depends on certain conditions, for example, the identity orientation owned by that company.

Organizational identity orientation theory is another frame used by former researchers to explain public CSR behavior [20,21]. Organizational identity orientation (OIO) is an extension of organizational identity theory and provides an integrative conceptual framework for understanding the link between a corporation and its contractual stakeholders [22]. OIO assumes that corporations have motivational differences when engaging in certain stakeholders; and it is these differences that influence how corporations focus on a disparate set of stakeholder claims [22]. If a company views itself primarily as being distinctive from its competitive counterparts, and keeps independent relation with its contractual stakeholders, it is individualistic-identity-oriented; if a company views itself primarily as a good partner of those with whom it interacts, and keeps dyadically interdependent relationship with its contractual stakeholders, it is relational-identity-oriented; yet if a company views itself primarily a good member to a larger community and works to improve the welfare of the community it values and/or belongs to, it is collectivistic-identity-oriented [22].

Yet, existing literature has not addressed the role of OIO in the relationships of organizational virtuousness and public CSR. Based to the analysis above, we posit that organizational identity orientation may moderate the relationship between organizational virtuousness and its public CSR behavior. Our research framework is shown in Figure 1. First, using empirical data from Chinese companies, we will examine the direct effect of organizational virtuousness on companies' public CSR behavior. Secondly, we will examine the moderating effect of organizational identity orientation between organizational virtuousness and companies' public CSR behavior.

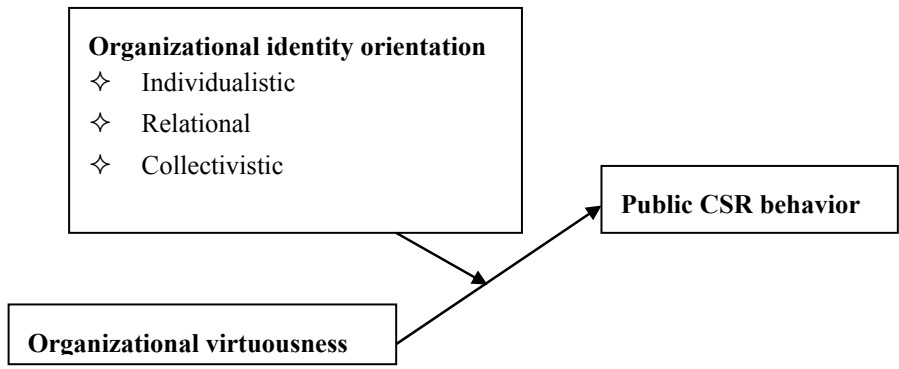

**Figure 1.** Research Model.

### 2.2. Hypothesis

The concepts of "virtue" and "virtuousness" both originate from 'vis-' which means power or strength in Latin, and ultimately from 'areté' which means excellence in Greek [23]. Although virtue ethics used to be understood as relevant only for individuals, some had justified the rationale and applied the concept to organizations [24,25]. Currently, it is widely accepted that both concepts can be used for analysis at both individual-level (i.e., personal virtuousness or virtue) and organizational-level (i.e., organizational virtuousness or virtue) [26]. In spite of this, virtue and virtuousness are distinctive constructs. The former is considered to be a property of individual or organizational character, whereas "virtuousness" is a construct associated with a constellation of virtues as an aggregate construct [27]. As such, organizational virtuousness refers to aggregates of organizational virtues, acting in combination, often manifested through processes, structures, attributes, and cultures in organizations, and in individual and collective actions [28]. Optimism, forgiveness, compassion,

trust, and integrity have been found to be those organizational virtues whose combination can capture the concept of organizational virtuousness [29]. Following the definition of Cameron and his colleagues, in our study we conceptualized organizational virtuousness as an aggregate and multidimensional construct including the aforementioned five virtues, optimism, trust, compassion, forgiveness, and integrity. We thus define virtuous corporation as one which possesses a system of stationery organizational virtues, thereby an ethos of virtuousness is acquired to provide proactive guidance for its behavior.

Three key defining attributes are associated with organizational virtuousness, namely human impact, moral goodness, and unconditional societal betterment [30]. Human impact determines that organizational virtuousness can help organizational members to control themselves and observe the principle of excellence, which is helpful for the moral development of organizational members [31]. Moral goodness implies that organizational virtuousness can direct the organization to pursue the "internal goods" or "goods of first intent", such as "love, wisdom, or fulfillment" as opposed to goods of second intent as "external goods", such as "profit, prestige, or power" which have an instrumental purpose [31]. Unconditional social betterment means organizational virtuousness will guide the organization to create benefit for others beyond self-interest without regard for reciprocity or reward, over and above mere participation in normatively prescribed volunteerism, which results in social betterment of the community [31].

In other words, virtuousness not only helps organizations prevent wrong doing, but enhances the likelihood of pursuing higher levels benefits for individuals and the society [17]. To this end, virtuous behavior is construed as being positively deviant to the extent that it militates against weaknesses, is counteractive to negative normative momenta that tend to reinforce conformance to accepted norms, and enables "extraordinariness" for pursuant of the "common good" [32]. Public CSR is just the so-called "societal benefit" and "common good" mentioned above. Therefore, we expect that:

**Hypothesis 1.** *Organizational virtuousness is positively associated with public CSR behavior.*

Although organizational virtuousness has a positive effect on public CSR behavior in general, this positive effect may be moderated by organizational identity orientation (OIO). The identity orientation construct was initially introduced for individual level of analysis [33,34]. Brickson [22] extended the construct to organizational level to describe the nature of relations between an organization and its immediate stakeholders as perceived by the organization's members. This construct answers the question of "who are we as an organization vis-a-vis our immediate stakeholders, as a sole entity, as a dyadic inter-entity relationship partner, or as a member of some larger collective" [22]. According to the locus of self-definition, three types of organizational identity orientation have been identified: (1) individualistic, (2) relational, and (3) collectivistic [22].

Individualistic-identity-oriented organizations or individualistic organizations define themselves as a sole entity, atomized and distinct from others [35]. This type of firms tend to generally forge relationships based on instrumentality and maintain relations to the extent that they enhance the organization's own aims such as uniqueness and profitability [35]. Because the self-definition is a sole entity, the ties between individualistic organizations and their contractual stakeholders tend to be weak and fluid [35]. For example, when necessary, in order to ensure their efficiency, individualistic organizations may change their partners (such as suppliers) at any time. Often times, taking public CSR may more or less harm the interests of their contractual stakeholders, at least for short-term benefits. Because the relationship is characterized by weak ties, when it comes to public CSR, they tend to act independently, less worrying about the interests or reactions of their contractual stakeholders. At the same time, because they are attributed to succeeding as individual entities for profitability, reputations, or market share, they may actively engage in public CSR activities to distinguish them from others and to maintain legitimacy as a responsible actor in a shared organizational environment [20]. Therefore, we expect that:

**Hypothesis 2a.** *Individualistic identity orientation will positively moderate the relationship between organizational virtuousness and public CSR behavior.*

On the contrary, relational-identity-oriented organizations or relational organizations define themselves as a dyadic relationship partner. This type of organizations tend to forge stakeholder relationships based on dyadic concern and trust, and observe a sense of responsibility to maintain a good relationship with salient stakeholders [20]. As opposed to viewing relationships as a means to an end, relational organizations view them largely as an end in itself [35]. Their self-view as interconnected to stakeholders through dyadic bonds produces a genuine desire to understand and benefit individual stakeholder. Thus, the ties between relational organizations and their contractual stakeholders are characterized by predominant strong dyadic ties [35]. Because of the attempt to maintain a strong tie with their immediate stakeholders, during business decisions, the relational organizations tend to concern about the well-being of particular outsiders and insiders with which their members perceive organizations having meaningful relationships. They strive to play a good partner role according to certain criteria provided by the organizations themselves or by a given stakeholder, or by both. Given that taking public CSR will more or less compromise the interests of contractual stakeholders, when those possessing strong relational identity orientation intend to take responsibility for the public, they are more likely worry about the interests and feelings of those stakeholders, which may reduce the commitment to public responsibility taking. From this, if a virtuous company is relational-identity-oriented, its willingness to take public CSR will become weaker. Therefore, we expect that:

**Hypothesis 2b.** *Relational identity orientation will negatively moderate the relationship between organizational virtuousness and public CSR behavior.*

Collectivistic-identity-oriented organizations or collectivistic organizations see themselves as a member of a larger group such as the society or community. This type of organizations will forge external and internal contractual stakeholder relationships based on a common goal. Although both collectivistic and individualistic companies view the relationship with immediate stakeholders as a means to an end, individualistic organizations use the relationship to meet their self-defined objectives, while collectivistic organizations use the relationship to meet common goals [35]. Additionally, unlike relational organizations emphasizing building close dyadic bonds, collectivistic firms view immediate stakeholder relationships as a means to promoting commonly held beliefs. Meanwhile, the collectivistic organizations focus on the protection and promotion of overall societal welfare and a strong motivation to contribute to the broader community [22]. As such, a firm with strong collectivistic identity orientation tends to maximize social interests rather than their own interests and has a strong incentive to take public CSR. Since a strongly collectivistic-identity-oriented company usually maintains a weak tie with contractual stakeholders, it will consider less of the interest of contractual stakeholders and less likely to be affected by them when making decisions on taking public responsibility. In view of this, if the virtuous company has a strong collectivistic identity orientation, it will be more active in taking public responsibility. Therefore, we expect that:

**Hypothesis 2c.** *Collectivistic identity orientation will positively moderate the relationship between organizational virtuousness and public CSR behavior.*

## 3. Method

### 3.1. Samples

This is an organizational-level study that needs to investigate a large number of corporations, so the task of sample collection was arduous. To this end, we asked the MBA students of a university

in Shanghai to help complete the survey. With their help, we successfully visited 88 firms where they work. These firms are mainly located in the Yangtze River Delta region in China and distributed in different industries. These companies included 31 state-owned firms, 37 privately-owned firms, and 20 foreign-invested firms. For a given firm in our sample, with the permission of top management, we randomly invited 3–12 employees to complete a questionnaire, which measured the firm's virtuousness, identity orientation, and public CSR. A total of 742 valid responses were received.

The demographic characteristic distribution of our sample is as following: regarding the gender, 44.3% was male, 55.7% was female; regarding the age, 26.3% were 25 years old and below, 35.3% were 26–30 years old, 19.9% were 31–35 years old, 7.8% were 36–40 years old,10.6% were 41 years old and up; regarding the education, junior high school and the bellow accounted for 2.2%, high school or technical secondary school accounted for 15.5%, college accounted for 39.6%, undergraduate accounted for 38%, graduate and up accounted for 4.7%; regarding the tenure with the current employer, 15% of them had less than 1 year, 22.1% had1–2years, 32.7% had2–5 years, 17.7% had 5–10 years, 12.5% had more than 10 years of experience in their organizations; regarding the rank, 71.8% were general staffs, 17.9% were supervisors, 9.3% were department managers, 0.9% were top managers.

### 3.2. Measures

**Organizational virtuousness:** Different from organizational virtue, which is a specific construct, organizational virtuousness is an abstract construct representing aggregates of organizational virtues. Thus, the data of organizational virtuousness cannot be attained directly, and it must be measured indirectly by aggregating those cardinal organizational virtues to a higher level. The 15-item scale proposed by Cameron et al. [29] was used to measure our independent variable, organizational virtuousness. In this scale, five cardinal virtues (optimism, trust, compassion, forgiveness, and integrity) are designed to represent the five dimensions of virtuousness (see Table 1). Responses were rated on five-point Likert scales ranging from 1 (strongly disagree) to 5 (strongly agree).

**Table 1.** Scale for Organizational Virtuousness.

| |
|---|
| Optimism |
| We are optimistic that we will succeed, even when faced with major challenges. |
| In this organization, we are dedicated to doing good in addition to doing well. |
| A sense of profound purpose is associated with what we do here. |
| Trust |
| Employees trust one another in this organization. |
| People are treated with courtesy, consideration, and respect in this organization. |
| People trust the leadership of this organization. |
| Compassion |
| Acts of compassion are common here. |
| This organization is characterized by many acts of concern and caring for other people. |
| Many stories of compassion and concern circulate among organization members. |
| Integrity |
| This organization demonstrates the highest levels of integrity. |
| This organization would be described as virtuous and honorable. |
| Honesty and trustworthiness are hallmarks of this organization. |
| Forgiveness |
| We try to learn from our mistakes here, consequently, missteps are quickly forgiven. |
| This is a forgiving, compassionate organization in which to work. |
| We have very high standards of performance, yet we forgive mistakes when they are acknowledged and corrected. |

**Organizational identity orientation:** Brickson [22] has developed a scale including three subscales to measure the three organizational identity orientations. Each subscale is composed of three items. With a sample of more than a thousand individuals in two industries, the convergent and discriminative validities of the three orientation constructs at the organizational level have been

tested [22]. This scale was adopted to measure our moderating variable (see Table 2). Responses were provided on five-point scales ranging from 1 (strongly disagree) to 5 (strongly agree).

**Table 2.** Scale for Organizational Identity Orientations.

| |
|---|
| Individualistic orientation |
| My organization views itself primarily as distinct and standing apart from other organizations. |
| What is most important to my organization is working to promote and maintain its own welfare (e.g., profitability, image, etc.). |
| My organization is most concerned about its distinctiveness from other organizations. |
| Relational orientation |
| My organization views itself primarily as a good partner to those with whom it interacts (e.g., employees, customers, nonprofit organizations). |
| What is most important to my organization is working to improve the welfare of particular others with whom the organization has significant and gratifying relationships (e.g., employees, customers, nonprofit organizations). |
| My organization is most concerned about its relationships with particular others whose welfare it values (e.g., employees, customers, a nonprofit organization). |
| Collectivistic orientation |
| My organization views itself primarily as a good member to a larger community (all those with whom it interacts, as a common group; a group of organizations promoting a cause it cares about; local community; etc.). |
| What is most important to my organization is working to improve the welfare of a community it values and/or belongs to (e.g., all those with whom it interacts, as a common group; a group of organizations promoting a cause; local community; etc.). |
| My organization is most concerned about its relationship with a greater community it values and/or belongs to (e.g., all those with whom it interacts, as a common group; a group of organizations promoting a cause it cares about; local community; etc.). |

**Public CSR Behavior:** Turker [36] has developed a scale including four dimensions to measure a company's social responsibility behavior. The first dimension is CSR to social and non-social stakeholders, involving such items like responsibility to society, responsibility to natural environment, responsibility to next generations, and responsibility to non-governmental organizations, etc. The remaining three dimensions are CSR to employees, CSR to customers, and CSR to government, respectively. Because the content included in the first dimension represent the common responsibilities of mankind, and they are considered to be ethical, philanthropic, and discretionary with their own intrinsic value, the first dimension just reflects the public CSR concept defined in this article. While the items included in other three dimensions represent the responsibility that a corporation should take for its contractual stakeholders, these three dimensions just embody the relational CSR concept defined in this article. Therefore, we will use the first dimension of Turker's scale to measure our dependent variable, public CSR behavior in this study. The first dimension includes 6 items (see Table 3). Responses were provided on five-point scales ranging from 1 (strongly disagree) to 5 (strongly agree).

**Table 3.** Scale for Public CSR Behavior.

| |
|---|
| Our company participates to the activities which aim to protect and improve the quality of the natural environment. |
| Our company makes investment to create a better life for the future generations. |
| Our company implements special programs to minimize its negative impact on the natural environment. |
| Our company targets a sustainable growth which considers to the future generations. |
| Our company supports the non-governmental organizations working in the problematic areas. |
| Our company contributes to the campaigns and projects that promote the well-being of the society. |

We also included control variables in corporate age and size. The literature indicated that larger and long-established firms are more likely to do better than smaller and younger firms in sustainable development. A possible reason is that larger and long-established firms are usually monitored more closely by the governments and the public, so they are more pressured to show better performance in sustainable development. On the other hand, larger and long-established firms may also have

more resources, such as financial and technological resources, to perform better [8]. In this study, corporate size was measured by the number of employees, and corporate age was measured by the operation life of the firms. Because the data of those two control variables is relatively large, logarithm transformation was taken before statistical analysis.

## 4. Results

### 4.1. Data Verification Analysis

**Organizational virtuousness:** A confirmatory factor analysis (CFA) using LISREL and maximum likelihood estimation was conducted on the multidimensional construct of organizational virtuousness in order to examine whether the second order model fit our data. Our findings confirmed the acceptable fit of our second order model ($\chi^2$ = 459.73, df = 83, RMSEA = 0.078, SRMR = 0.032, NNFI = 0.98, CFI = 0.99, GFI = 0.92). All items loaded significantly on their respective factor with standardized loadings ranging from 0.75 to 0.86. Moreover, the five factors loaded significantly on their higher order factor, and their standardized loadings ranged from 0.81 to 0.94. The result shows that the validity of our scale is satisfactory, and it is reasonable to composite the five virtues to a high order construct (virtuousness). So, in the following statistical analysis, we will average the 15 items to obtain a measured value for organizational virtuousness. Meanwhile, each dimension of organizational virtuousness as well as the construct of organizational virtuousness as a whole had satisfactory reliability as Cronbach $\alpha$ surpassed 0.70 (optimism = 0.8874, trust = 0.8860, Compassion = 0.8710, integrity = 0.9027, forgiveness = 0.8741, and organizational virtuousness = 0.9594).

**OIO:** In order to test the validity of this instrument, a CFA (using LISREL and maximum likelihood estimation) was carried on the three-factor model. The result showed that the three-factor model fit the data acceptably ($\chi^2$ = 380.71, df = 24, RMSEA = 0.086, SRMR = 0.046, NNFI = 0.94, CFI = 0.96, GFI = 0.90). All items had satisfactory item reliability as their standardized loadings on their respective factor ranged from 0.65 to 0.95. Meanwhile, each subscale had satisfactory reliability as Cronbach $\alpha$ surpassed 0.70, the subscale for individualistic identity orientation = 0.7515, the subscale for relational identity orientation = 0.8513, the subscale for collectivistic identity orientation = 0.7758.

**Public CSR Behavior:** In order to test the validity of this instrument, a CFA (using LISREL and maximum likelihood estimation) was conducted on the one-factor model. The results showed that the one-factor model fit our data acceptably ($\chi^2$ = 390.38, df = 34, RMSEA = 0.097, SRMR = 0.051, NNFI = 0.94, CFI = 0.95, GFI = 0.90). Meanwhile, this scale had satisfactory reliability as Cronbach $\alpha$ is 0.8986.

### 4.2. Aggregate Analyses

Our model was focused on organization-level as the unit of analysis. Specifically, we investigated three constructs: organizational virtuousness, organizational identity orientation, and public CSR behavior at the organization level. Thus, each of these three constructs is a property of the organizations themselves, not the individuals that compromise the organizations. Yet, as we were unable to obtain data for the organization-level constructs directly, individual members in organizations became the actual source of data. Therefore, we need to aggregate the individual-level data to organizational level after justifying within-group agreement ($r_{wg}$) and intra class correlation coefficient ICC (1) and ICC (2). Generally speaking, only when the median or mean of $r_{wg}$ is greater than 0.70, the ICC (1) is less than 0.5 and F test is significant, the ICC (2) is over 0.7, can we aggregate the individual-level data to an upper level [37].

Taking the responses from the same company as a group, we divided the sample into 88 groups. We first computed $r_{wg}$ statistic for 3 variables for each group. The result showed that for public CSR behavior, the average of the 88 groups' $r_{wg}$ was 0.90 and the median was 0.94; for organizational virtuousness, the average of $r_{wg}$ was 0.92 and the median was 0.9768; for individualistic identity orientation, the average of $r_{wg}$ was 0.8234 and the median was 0.8704; for relational identity orientation,

the average of $r_{wg}$ was 0.8266, the median was 0.9009; for collectivistic identity orientation, the average of $r_{wg}$ was 0.8490, the median was 0.8845.

We then conducted a one-way ANOVA and computed the ICC (1) and ICC (2) statistics for the three variables based on the 88 groups. The result showed that for public CSR behavior, the ICC (1) was 0.4157, the ICC (2) is 0.8571; for organizational virtuousness, the ICC (1) was 0.4579, the ICC (2) is 0.8776; for individualistic identity orientation, the ICC (1) was 0.3726, the ICC (2) is 0.8335; for relational identity orientation, ICC (1) was 0.3924, the ICC (2) is 0.8448; for collectivistic identity orientation, ICC (1) was 0.3934, the ICC (2) is 0.8454.

In short, the results of aggregate analyses showed that it was acceptable to aggregate the individual-level data for the variables to a collective level. With the acceptable group level data, we proceed to conducting an organization-level analysis for testing our hypothesis.

### 4.3. Descriptive Statistics and Correlation Analysis

The result of descriptive statistics and correlation analysis for research variables and control variables was shown in the following Table 4.

**Table 4.** Means, Standard Deviations and Correlations.

| Variables | | Mean | SD | 1 | 2 | 3 | 4 | 5 | 6 | 7 |
|---|---|---|---|---|---|---|---|---|---|---|
| 1. corporation age | | 2.470 | 0.898 | 1 | | | | | | |
| 2. corporation size | | 2.572 | 0.758 | 0.516 ** | 1 | | | | | |
| 3. organizational virtuousness | | 3.890 | 0.597 | −0.117 | −0.075 | 1 | | | | |
| Organizational identity orientation | 4. individualistic | 3.631 | 0.503 | −0.196 | 0.010 | 0.692 ** | 1 | | | |
| | 5. relational | 3.910 | 0.594 | −0.054 | 0.015 | 0.838 ** | 0.744 ** | 1 | | |
| | 6. collectivistic | 3.918 | 0.551 | 0.000 | 0.162 | 0.742 ** | 0.804 ** | 0.833 ** | 1 | |
| 7. Public CSR behavior | | 3.757 | 0.634 | 0.013 | 0.172 | 0.726 ** | 0.557 ** | 0.636 ** | 0.647 ** | 1 |

Note: $n = 88$; ** $p < 0.01$, * $p < 0.05$.

### 4.4. Hypothesis Testing

Harman's single-factor test was used in this study to evaluate the influence of common method variance (CMV) on the results of statistical analysis. Using the 742 individual-level data, we loaded all of the variables in the study into an exploratory factor analysis (EFA) and examined the unrotated factor solution to determine the number of factors that are necessary to account for the variance in the variables. The result of EFA showed, according to the principle of eigenvalue greater than 1, five factors emerged from the factor analysis, their variance contribution rates were 51.141%, 6.936%, 4.529%, 3.370%, and 3.000%, respectively. So, results of single-factor test suggest that common-method bias is not a concern for this study.

Taking public CSR behavior as the outcome variables, hierarchical regression analysis was conducted to test the moderating effect of three types of organizational identity orientation on the relationship between organizational virtuousness and public CSR behavior. The results were reported in Table 5.

Model 1 was used to test the effect of control variables on the outcome variable. The results showed that corporation age was negatively related to public CSR behavior, corporation size is positively related with public CSR behavior, yet both effects are not significant. In Model 2, we added both independent and the outcome variables. The results showed that organizational virtuousness was positively associated with public CSR behavior and the influence is significant (Beta = 0.741). Thus, Hypothesis 1 was supported.

Model 3 was to test the effect of moderating variables on the outcome variable. The results showed that relational organizational identity orientation was negatively correlated with public CSR behavior, collectivistic organizational identity is positively correlated with public CSR behavior, and individualistic organizational identity orientation has no relationship with public CSR behavior, yet

all the three types of effect are not significant. Model 4 was used to test the moderating effect of organizational identity between independent variable and result variable. The results showed that individualistic organizational identity orientation positively moderated the effect of organizational virtuousness on public CSR behavior, the moderating effect was significant (Beta = 0.280), the interactive effect is shown in Figure 2, Hypothesis 2a was supported; relational organizational identity orientation negatively moderates the effect of organizational virtuousness on public CSR behavior, the moderating effect was significant (Beta = −0.622), the interactive effect was shown in Figure 3. Therefore, Hypothesis 2b has been supported; collectivistic organizational identity orientation positively moderates the effect of organizational virtuousness on public CSR behavior, the moderating effect is significant (Beta = 0.405), the interactive effect is shown in Figure 4, so Hypothesis 2c has been supported.

**Table 5.** Result of Hierarchical Regression.

| Model | | Model 1 | Model 2 | Model 3 | Model 4 |
|---|---|---|---|---|---|
| Predictors and Outcomes | | Public CSR | Public CSR | Public CSR | Public CSR |
| | | Beta | Beta | Beta | Beta |
| Corporation age | | −0.103 | −0.024 | −0.017 | −0.001 |
| Corporation size | | 0.225 | 0.240 ** | 0.202 * | 0.154 |
| Organizational virtuousness | | | 0.741 *** | 0.674 *** | 0.670 *** |
| Organizational identity orientation | Individualistic | | | 0.000 | −0.086 |
| | Relational | | | −0.091 | −0.235 |
| | Collectivistic | | | 0.189 | 0.365 ** |
| Organizational virtuousness × Individualistic identity orientation | | | | | 0.280 * |
| Organizational virtuousness × Relational identity orientation | | | | | −0.622 *** |
| Organizational virtuousness × Collectivistic identity orientation | | | | | 0.405 ** |
| F | | 1.640 | 38.397 | 19.303 | 19.412 |
| F significance | | 0.200 | 0.000 | 0.000 | 0.000 |
| Adjusted R2 | | 0.015 | 0.563 | 0.558 | 0.656 |
| ΔR2 | | 0.037 | 0.541 | 0.010 | 0.103 |
| ΔF significance | | 0.200 | 0.000 | 0.575 | 0.000 |

Note: * $p < 0.05$; ** $p < 0.01$; *** $p < 0.001$.

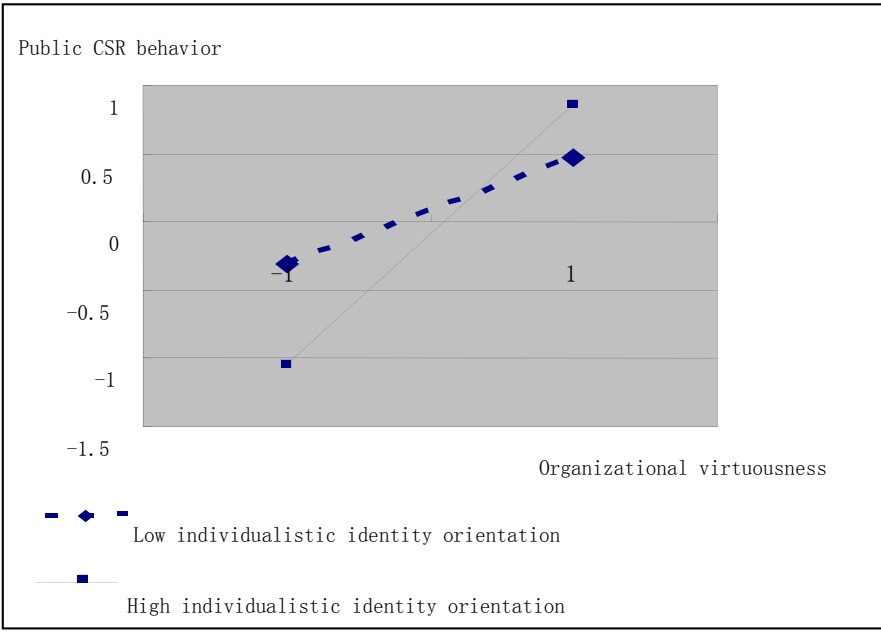

**Figure 2.** The Interactive Effect for Hypothesis 2a (Beta = 0.28).

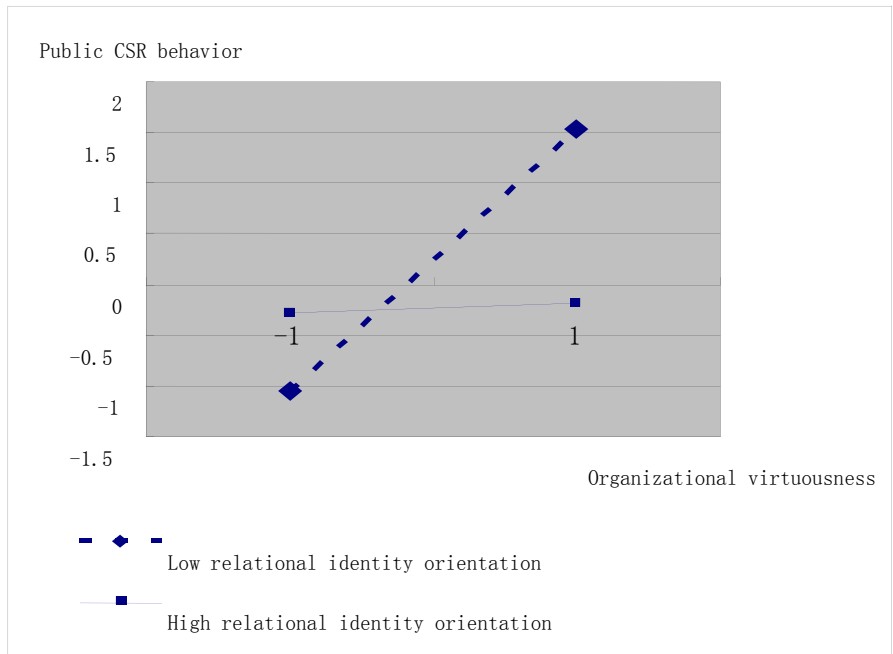

**Figure 3.** The Interactive Effect for Hypothesis 2b (Beta = 0.622).

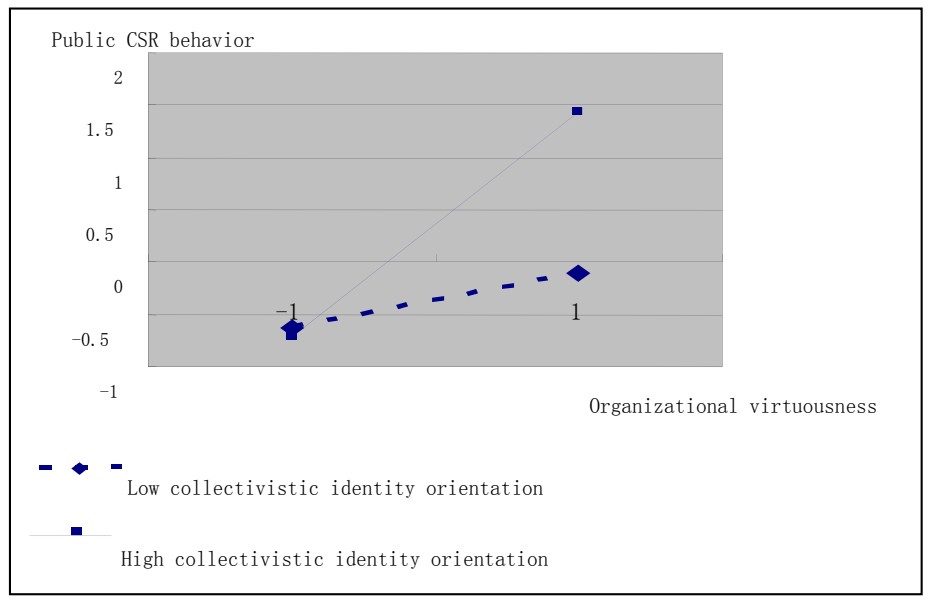

**Figure 4.** The Interactive Effect for Hypothesis 2c (Beta = 0.405).

## 5. Discussion

### 5.1. Conclusions

The existing literature only mentioned that organizational virtuousness results in genuine responsible behavior, but not touched the following question "would virtuous corporations be more willing to take public CSR than others?" Our study showed that organizational virtuousness is positively related to public CSR, but this positive effect may be amplified or buffered by another important organizational trait, namely organizational identity orientation. Specifically speaking, the individualistic and collectivistic identity orientation has an amplifying function, while the relational identity orientation has a buffering function. So, the answer to the above question is that virtuous corporation is not necessarily more willing to take public CSR, to some degree it depends on the type of identity orientation possessed by the virtuous corporation.

This conclusion can be illustrated by an example. On 12 May 2008, an earthquake struck Wenchuan, China, resulting in serious casualties and properties damages. During the week immediately after the earthquake, two local private firms reacted with their respective CSR behavior that stirred completely different public responses. On the day of the earthquake, Vanke Group, the largest real estate builder in China, immediately announced the donation of a disaster relief assistance fund, amounting to 2 million RMB. A week later, JDB Group, a soft-drink bottling firm producing herb teas and soft drinks, announced a donation of RMB 100 million for disaster relief and construction. Yet, comparing to the amount donated by JDB Group, the amount by Vanke Group was sarcastically criticized by millions of netizens, generating a wide range of negative publicity throughout the Chinese internet social media. In contrast, owing to its generosity and subsequent favorable publicity, JDB Group's market share grew to the leading position in the national filling beverage market in the following years. In responding to the criticism, the Chairman of Vanke Group, Mr. Shi Wang argued that 2 million RMB was an appropriate amount for their company to donate. Firms ought to contribute (to the society) continuously, and the donations should not be obliged as burdens for employees and corporations. Charitable donation should not become a contest among corporations. However, Wang Shi's explanation undoubtedly fueled the public's criticism and immediately attracted more fierce attacks on Vanke Group and Wang Shi himself. So, Wang Shi had to make open apologies to the public on a TV show and then the company issued a "supplementary donation" announcement, with an additional donation of RMB 100 million for post-disaster reconstruction. So far, the so-called "donation-gate" crisis has come to an end [38].

For quite some time, Wang Shi, known as "good citizen", enjoys a high reputation in China's real estate industry. Vanke Group, under his leadership, has been named "the best corporate citizen in China" for many years [38]. The public has no doubt about its donation genuineness and believe that it is a virtuous corporation. Yet, although the donation was sincere, Vanke Group's willingness to contribute to the disaster-stricken areas was weaker than that of other corporations, because Vanke group takes the interests of its contractual stakeholders (shareholders and employees) more into account in the decision-making process of the first donation. This case shows that a virtuous corporation may not be necessarily more willingness to take public CSR than other counterparts, and the intention and willingness to take public CSR are likely to be affected by how it perceives their relationships with its contractual stakeholders.

Now, we can explain the embarrassment faced by Vanke Group and its CEO Mr. Shi Wang. Needless to say, Vanke Group has a high level of organizational virtuousness. However, Vanke Group is a highly relational-identity-oriented company. It is just because of these two organizational traits possessed simultaneously by Vanke Group that put the company into a passive position in the so-called "donation-gate" crisis. Specifically speaking, donations for the disaster areas are corporate responsibility for the public. However, this kind of charitable contributions may affect more or less the interests of contractual (or immediate) stakeholders. Because the virtuous Vanke Group took the interests of contractual stakeholders into consideration in the process of donation, so the company's commitment to public responsibility has declined, which caused a public outcry. On the contrary, if the virtuous Vanke Group is individualistic-identity-oriented or collectivistic-identity-oriented, it would not have been in such an awkward position, because the latter two types of companies tend to act independently in the process of public CSR taking, less worrying about the interests or reactions of their contractual stakeholders.

## 5.2. Managerial Implications

Corporations are cells of a society, and the society is the source of a corporation's benefit, companies should put back into the society what they have taken out. Then, how to improve the enthusiasm of a company to take public CSR?

The results of this study show that, in general, organizational virtuousness has a positive impact on public CSR behavior. So, to promote a company's motivation in taking public CSR, it is critical

to cultivate organizational virtuousness. Moore [39] suggested that, to build a virtuous company, designing an internal governance system that tends to crowd in rather than crowd out virtue for the company is the most important thing to do. This kind of internal governance system has the following eight parameters [39]. At first, the business objectives should contribute to the overriding good of the community; the second, hiring the employees and agents with pro-social intrinsic preferences; the third, intrinsic motivation should be considered in to the greatest extent possible during job design; the fourth, executive pay should be curbed, largely fixed and fair salaries should become the norm; the fifth, decision-making processes are designed to strengthen participation and self-governance; the sixth, managers must cultivate trust in employees; the seventh, encouraging group identity and ensuring that all employees feel part of the in-group; the eighth, organizational transparency (internal and external) is paramount.

However, to promote a company's motive to take public CSR, it is far from enough to rely solely on the cultivating of organizational virtuousness. Because the results of this study also showed that whether a virtuous company is willing to contribute more to the society depends to some degree on the identity orientation owned by that company. Therefore, it is necessary for the government to manage companies based on their identity orientation. Brickson [35] pointed out individualistic-identity-oriented organizations have an instrumental motivation to take public CSR in order to improve corporate reputation and uniqueness. Hence, for this type of companies, the government should guide and encourage them to take public CSR, set them up as a model enterprise, and award them to improve their social reputation. Relational-identity-oriented organizations tend to consider the interests of contractual stakeholders ahead of the public interest. Hence, for this type of companies, local government should restrain and regulate their deviation behavior, and remind them that the public interest or social interest is the first one in front of many stakeholders. Because the mission and purpose of collectivistic-identity-oriented organizations is to pursue the maximal interest of the whole society, their motives to take public CSR usually are unconditional. For this type of companies, the government should support and assist them to serve the community, and strive to build good platform for them to take more public responsibility.

*5.3. Theoretical Implications*

This article tentatively discussed why a company is willing to take public CSR from trait approach. It was an entirely new exploration about the motivation of corporations' public CSR initiative.

Although instrumental, normative, and moral approaches have been widely used by former scholars to explain the motivation of a corporation to take public CSR, there are still some shortcomings. For example, Atkinson & Galaskiewicz [40] argued that the so-called economic rationality of public CSR behavior is not very reliable, and there is no evidence that public CSR behavior is to expand sales, improve public relations, or enhance the image of the corporation. Moreover, confronted with the temptation of economic interests, the moralization based on social contract becomes very feeble to impact the philanthropic behavior of a company. Although the normal approach has strong explanatory power for a corporation's public CSR initiative, this approach only emphasizes the extrinsic motivation, namely a company's public CSR activities are just passive response to the external institutional system.

However, trait approach is different from the above three approaches. At first, both instrumental and moral approach emphasize the influence of behavioral environment (institutional norms for the former, social moral norms for the later) on a company's public CSR motivation, instrumental approach emphasizes the influence of behavioral outcomes on a company's public CSR motivation, while trait approach focuses on the influence of organizational character traits on companies' public CSR motivation. Secondly, instrumental, normal, and moral approaches deal with company' extrinsic motivation to take public CSR, while trait approach deal with company's intrinsic motivation to take public CSR. So, our study was a pioneering research which has important theoretical value.

*5.4. Limitations*

This study has empirically examined the interactive effects of organizational virtuousness and organizational identity orientation on public CSR behavior. We took a new approach to addressing the motivation of organizations to taka public CSR. Yet, this study may have the following limitations.

Firstly, our study could be criticized for the small sample size. Both the level of theory and the level of data analysis are at the organization level in our study, and it is the organization as a whole not an individual member in the organization represents a research sample. So, a lot of companies should be investigated as samples in our study.

Secondly, the main research variables in our study are organization-level variable which cannot be measured directly. A common practice is, selecting some employees randomly from each company, investigating these employees' perception about the above variables, under the condition that the internal consistency among individual perceptions in each company meet the requirement, the individual-level data is integrated into the organization-level by means of the average. We only drew 3–12 individual members from each company to represent that company in current study, but we think more employees should be drawn from a company in future studies.

Thirdly, the effect of common method bias on results should be alleviated. Collecting data from different sources can alleviate the aforementioned concern. For example, future studies can collect the data on organizational virtuousness from employee and collect the data on organizational identity orientation and public CSR behavior from outsiders, such as suppliers, customers, and so on. Also, more objective measurement should be used.

*5.5. Future Directions*

As mentioned above, corporate social responsibility includes relational CSR and public CSR [5]. This article only focused on corporation's public responsibility behavior towards society or community. Future research can use the relational CSR as outcome variables to verify the moderating effect of organizational identity orientation between organizational virtuousness and relational CSR.

This article has verified the positive effect of organizational virtuousness on public CSR and emphasized the importance of cultivating organizational virtuousness. Therefore, future research should explore the antecedent variables of organizational virtuousness. Searle, &Barbuto [41] proposed that servant leadership which characterized by organizational stewardship and altruistic calling will be related to the development of organizational virtuousness. Organizational stewardship is the extent to which a servant leader prepares his or her organization to make a positive contribution to society [42]. Altruistic calling describes the servant leader's deep-rooted desire to make a positive difference in the lives of all stakeholders and is the foundation of the construct [42]. Therefore, future research can explore the effect of servant leadership on organizational virtuousness.

Finally, the relational-identity-oriented organization is very concerned about the relationship maintenance with its immediate stakeholders and tries its best to play a good partner role before those stakeholders. Yet, those stakeholder groups usually have different even conflict interest claims. So, the relational-identity-oriented organization will encounter role conflict when it balances the interests of those stakeholders or makes trade-offs among those stakeholders. Future research should focus on how to reduce the role conflict encountered by relational identity-oriented organizations.

**Author Contributions:** Y.L. was responsible for theoretical model building, study design, data collection, manuscript writing and revision in this study. G.G.W. was responsible for manuscript revision and English polishing in this study. Y.C. was responsible for data statistics and contribution in this study.

**Funding:** This research was funded by The National Social Science Fund of China (Grant No. 15BSH103) and The Plateau Discipline of Business Administration Construction Fund of Shanghai Lixin University of Accounting and Finance.

**Conflicts of Interest:** The authors declare no conflict of interest.

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
