# Peer review of "Why Are Corporations Willing to Take on Public CSR? An Organizational Traits Approach"

_sustainability, doi:10.3390/su11020524_

Round 1

Reviewer 1 Report

Abstract 

Authors use term 'Corporation social responsibility' I would suggest to check the whole paper according to correct it or stay consistant with the regular name of the concept: "Corporate Social Responsibility". 

In my opinion the novelty in this paper lies on the examining the effect of OIO on the CSR Philantropy. In Abstract I would advise to resign from the first sentence used in the abstract and in general rethink the conceptualization of CSR into two Public and relational. In my opinion it is a big shortage, lacking substantial theory behind that may mislead the readers. Maybe it will be sufficient to explain more in details the theoretical perspectives on CSR (instrumental, normative and moral). Or - if authors decide to keep the relational-public distinction they sthould show the mutual interrelation between these different perspectives. 

Instroduction 

Language requires correction (eg. "are more able to sense areas of need following a disaster”; Vanke Co. Or Wanke; that a virtuous 47 corporation may not be necessarily more willingness to take public CSR than others.)

The aim: 

line 55-56 „First, we differentiated corporation social responsibility into the relational responsibility for the stakeholders (relational CSR) and the public responsibility for the whole society (public CSR).” How is this concept linked with the OIO (Brickson 2005). In the theoretical framework authors use a range of different concepts and theories, but the paper lack on the substantive, consistent and clear linkages between these concepts. Maybe authors could propose a graphic for the theoretical framework and more privide more specific explainations of the whole framwork? For example OIO ist not just theoretical framework for explaining CSR behavior. It may be used with this regard but authors should make clear what do they understand and why they think researching on this relationship may be important. The paper lack the required clarity, because authors indicate different concepts but don't provide substantial and specific conclusions on what they are working with and why. To assume here a specific research problem should be stated. 

The explaination of the case (Vanke) at the beginning of the article is not clearly described how it is linked to the organizational identity orientation 

"We demonstrated the distinct effect of organizational virtuousness on 61 corporation’s public CSR behavior and the moderating effect of organizational identity orientation. " THis clue should be reflected in the literature review. So the problem is [1] what impacts CSR public [in my opinion authors understood this concept as philantropic CSR], what kind of motives stay behind it and OIO may help to make this point clear. 

 „Third, our results suggest that organizational virtuousness is a sufficient but not necessary condition 63 for a company’s intention to take on public CSR, and that intention also depends on the type of 64identity orientation possessed by the virtuous company. Finally, this study reveals the motivation of 65 corporations to take public CSR from the perspective of organizational traits with evidence and 66 provide guidelines for companies to engage in CSR initiatives more effectively.”  - This is rather information suitable in the abstract than in the Introduction

Literature review

CSR from definition is oriented towards stakeholder and political CSR may be understood as directed towards publicity, media in general public opinion. The differentiation in relational and public CSR is very vague concept and lack of theoretical foundations. Authors omit the large literature on the instrumental and normative aspects of stakeholder theory. I would suggest to look for better evidence to conceplualize the CSR. 

"The relational CSR is more or less transactional, 87 instrumental, and compulsory, whereas the public CSR is in the ethical, philanthropic, and 88 discretional domain. "

This sentence is argueable and authors treat it as a thoeretical foundation without providing any sufficient support.  

Similarly Authors don’t provide support in the literature review for the statment that: However, the existing literature hasn’t covered how to promote a corporation’s motivation to engage 92 in disaster relief donation and how societal effects impact on corporation’s CSR decision-making. 93 There is a research gap we aim to fill in this paper.”

THere is literature on the motives of CSR philantropy, what drives CSR philantropy. Authors omit the fact.

Theoretical model 

TM should be better connected to the literature review. I would suggest to restructure the both acapist and take more systemic view. 

Measures

Scale – the information on the scale testing could be supported by some data and deeper explaination. The orginal version of the scale on OIO (Brickson 2005) is rather qualitative in nature. If authors were using this scale and have transponded it into a quantitative one some deeper explaination should be considered. I would expect more details in scale explaination, not just "sample items". 

This research is done on human aspect (perception) and authors should refer to the social desirability aspects and explain how they deal with it.

Discussion 

Should be linked to the literature review (which hasn't been done properly). Duscussion part is to improve

General

In general I would recommend to rethink the problem and the results. Authors were asking respondents about their perceptions on OIO and CSR Public Behavior. They were not collecting the data on sponsorships and philantropic funds, but the perception of it. This is a difference and it recuires a clear explaination that this study is not on CSR Public Behavior, but its perception.

To summarize the comments I would recommend:

1.    Making the clear poing about the research. In my opinion authors are testing the relationship between the perceived OIO and perceived CSR Philantropy. After reading the paper I’m not convinced that the concept of organizational virtuosness is here the core problem. Maybe authros should think about changing the narratives or changing the model tested. 

2.    Rewrite the literature review and refer to the broad range of studies done on the motives for CSR philantropy. Here also a range of stududies should help authors to find better evidences for hypothehses justifications.

3.     Strenghten the hypothesis justifications

4. Explain the measures and its testing

5.  Explaining how they did they deal with the social desirability aspects

6.    Provide the discussion that refers to the existing research on the OIO and Corporate Philantropy 

Author Response

Abstract

Authors use term 'Corporation social responsibility' I would suggest to check the whole paper according to correct it or stay consistant with the regular name of the concept: "Corporate Social Responsibility".

Response: Amendments have been made as required.

In my opinion the novelty in this paper lies on the examining the effect of OIO on the CSR Philantropy. In Abstract I would advise to resign from the first sentence used in the abstract and in general rethink the conceptualization of CSR into two Public and relational. In my opinion it is a big shortage, lacking substantial theory behind that may mislead the readers. Maybe it will be sufficient to explain more in details the theoretical perspectives on CSR (instrumental, normative and moral). Or - if authors decide to keep the relational-public distinction they sthould show the mutual interrelation between these different perspectives. 

Response: We decide to keep the relational-public distinction, the literature support has been provided in this paper. For more details, please refer to the relevant parts of the paper.

Instroduction

Language requires correction (eg. "are more able to sense areas of need following a disaster”; Vanke Co. Or Wanke; that a virtuous 47 corporation may not be necessarily more willingness to take public CSR than others.)

Response:  Amendments have been made as required.

The aim:

line 55-56 „First, we differentiated corporation social responsibility into the relational responsibility for the stakeholders (relational CSR) and the public responsibility for the whole society (public CSR).” How is this concept linked with the OIO (Brickson 2005). In the theoretical framework authors use a range of different concepts and theories, but the paper lack on the substantive, consistent and clear linkages between these concepts. Maybe authors could propose a graphic for the theoretical framework and more privide more specific explainations of the whole framwork? For example OIO ist not just theoretical framework for explaining CSR behavior. It may be used with this regard but authors should make clear what do they understand and why they think researching on this relationship may be important. The paper lack the required clarity, because authors indicate different concepts but don't provide substantial and specific conclusions on what they are working with and why. To assume here a specific research problem should be stated.

Response: We have reorganized the logical structure of the paper.

The explaination of the case (Vanke) at the beginning of the article is not clearly described how it is linked to the organizational identity orientation 

Response: We have reorganized the logical structure of the paper.

"We demonstrated the distinct effect of organizational virtuousness on 61 corporation’s public CSR behavior and the moderating effect of organizational identity orientation. " THis clue should be reflected in the literature review. So the problem is [1] what impacts CSR public [in my opinion authors understood this concept as philantropic CSR], what kind of motives stay behind it and OIO may help to make this point clear. 

Response: We have reorganized the logical structure of the paper.

 „Third, our results suggest that organizational virtuousness is a sufficient but not necessary condition 63 for a company’s intention to take on public CSR, and that intention also depends on the type of 64identity orientation possessed by the virtuous company. Finally, this study reveals the motivation of 65 corporations to take public CSR from the perspective of organizational traits with evidence and 66 provide guidelines for companies to engage in CSR initiatives more effectively.”  - This is rather information suitable in the abstract than in the Introduction

Response: We have reorganized the logical structure of the paper.

Literature review

CSR from definition is oriented towards stakeholder and political CSR may be understood as directed towards publicity, media in general public opinion. The differentiation in relational and public CSR is very vague concept and lack of theoretical foundations. Authors omit the large literature on the instrumental and normative aspects of stakeholder theory. I would suggest to look for better evidence to conceplualize the CSR. 

Response: We have distinguished the concept between relational CSR and public CSR. For more details, please refer to line 21-36.

"The relational CSR is more or less transactional, instrumental, and compulsory, whereas the public CSR is in the ethical, philanthropic, and 88 discretional domain. "This sentence is argueable and authors treat it as a thoeretical foundation without providing any sufficient support.  

Response: We provide literature support for this statement. For more details, please refer to line 35.

Similarly Authors don’t provide support in the literature review for the statment that: However, the existing literature hasn’t covered how to promote a corporation’s motivation to engage 92 in disaster relief donation and how societal effects impact on corporation’s CSR decision-making. 93 There is a research gap we aim to fill in this paper.”

THere is literature on the motives of CSR philantropy, what drives CSR philantropy. Authors omit the fact.

Response: We have reorganized the logical structure of the paper.

Theoretical model

TM should be better connected to the literature review. I would suggest to restructure the both acapist and take more systemic view. 

Response: Amendments have been made as required. For more details, please refer to line 48-70.

Measures

Scale – the information on the scale testing could be supported by some data and deeper explaination. The orginal version of the scale on OIO (Brickson 2005) is rather qualitative in nature. If authors were using this scale and have transponded it into a quantitative one some deeper explaination should be considered. I would expect more details in scale explaination, not just "sample items". 

This research is done on human aspect (perception) and authors should refer to the social desirability aspects and explain how they deal with it.

Response: Amendments have been made as required. For more details, please refer to line 239-262.

Discussion Should be linked to the literature review (which hasn't been done properly). Duscussion part is to improve

Response: Amendments have been made as required. For more details, please refer to line 368.

General

In general I would recommend to rethink the problem and the results. Authors were asking respondents about their perceptions on OIO and CSR Public Behavior. They were not collecting the data on sponsorships and philantropic funds, but the perception of it. This is a difference and it recuires a clear explaination that this study is not on CSR Public Behavior, but its perception.

Response: The main research variables in our study are organization-level variable which cannot be measured directly. In OB field, a common practice is, selecting some employees randomly from each company, investigating these employees’ perception about the above variables, under the condition that the internal consistency among individual perceptions in each company meet the requirement, the individual-level data is integrated into the organization-level by means of the average. Future studies can collect the data on organizational virtuousness from employee, and collect the data on organizational identity orientation and public OCB behavior from outsiders, such as suppliers, customers, and so on.  Also, more objective measurement should be used.

To summarize the comments I would recommend:

1. Making the clear poing about the research. In my opinion authors are testing the relationship between the perceived OIO and perceived CSR Philantropy. After reading the paper I’m not convinced that the concept of organizational virtuosness is here the core problem. Maybe authros should think about changing the narratives or changing the model tested. 

Response1: We have reorganized the logical structure of the paper.

2. Rewrite the literature review and refer to the broad range of studies done on the motives for CSR philantropy. Here also a range of stududies should help authors to find better evidences for hypothehses justifications.

Response2: Amendments have been made as required.

3. Strenghten the hypothesis justifications

Response3: Amendments have been made as required.

4. Explain the measures and its testing

Response4: Amendments have been made as required.

5. Explaining how they did they deal with the social desirability aspects

Response5: Amendments have been made as required.

6. Provide the discussion that refers to the existing research on the OIO and Corporate Philantropy.

Response6:  Amendments have been made as required. 

Reviewer 2 Report

Thank you for the opportunity to review your paper about the analysis of the relation between organizational virtuousness and public CSR.

There are several strengths to your study and some limitations which would need to be addressed; I’ll confine these observations to the following headings.

Originality

The paper contributes to the existing research, as limited attention has been given to the differentiation of CSR into relational and public.

Relationship to the literature

The literature review should be improved, including some more references about CSR. It would be useful for a better understanding, to cite the main studies explaining the meaning of corporate social responsibility.

Methodology

With specific regard to the empirical study illustrated in the paper, there are some methodological limitations.

Much more detail is needed about your research process. For example:

How did you select the empirical study’s companies? What documents were accessed? How did you provid the questionnaire to the selected employees? Over what time period?

Some research questions are included in section 3 rather than in the introduction; I suggest to underline all the research questions in the introduction.

Language

The manuscript contains some spelling errors.

I hope you find these comments helpful for furthering your work.

Author Response

Thank you for the opportunity to review your paper about the analysis of the relation between organizational virtuousness and public CSR.

There are several strengths to your study and some limitations which would need to be addressed; I’ll confine these observations to the following headings.

Originality

The paper contributes to the existing research, as limited attention has been given to the differentiation of CSR into relational and public.

Response: Thanks

Relationship to the literature

The literature review should be improved, including some more references about CSR. It would be useful for a better understanding, to cite the main studies explaining the meaning of corporate social responsibility.

Response: Some references have been added.

Methodology

With specific regard to the empirical study illustrated in the paper, there are some methodological limitations.

Much more detail is needed about your research process. For example: How did you select the empirical study’s companies? What documents were accessed? How did you provid the questionnaire to the selected employees? Over what time period?

Response: Amendments have been made as required. For more details, please refer to line 214-221.

Some research questions are included in section 3 rather than in the introduction; I suggest to underline all the research questions in the introduction.

Response: Amendments have been made as required. The paper has been reconstructed.

Language

The manuscript contains some spelling errors.

Response: Spelling errors have been corrected.

I hope you find these comments helpful for furthering your work.

Response: Your suggestion has helped me a lot. Thank you.

Reviewer 3 Report

The effect of organizational virtuousness on CSR is worthy to examine. I liked the tone of the paper, although there are a few questions that I would like the authors to clarify, and, hopefully, with this exercise, they may decide to make some changes in their paper.

1. Sufficient and necessary conditions? In several moments of their text they mention that "organizational virtuousness" (OV) is a sufficient but not necessary condition for Public CSR (PCSR).  Based on the Research Model (Figure 1) they want to investigate the relation between OV and PCSR, and they include as "moderator factors" the three types of Organizational Identity Orientation (OIO). To my understanding, they want to test that OV is not enough in order to explain why a company engages or succeeds in PCSR, but it is also necessary to include other variables such as OIO. Therefore, what the models says is that OV is necessary but not sufficient, right? If I am right, then you need to review several sentences along the text (for example, abstract, or p. 12, lines 451-454).

2. Does Public CSR mean engaging in disaster relief?. There is a confusing use of the concept of Public CSR along the text. In certain moments, the authors seem to refer to PCSR as how the companies contribute to sustainable development (for example, p2, line 83; or p.5, line 196: "Public CSR is just the so-called "societal benefit" and "common good" mentioned above"), but in other moments they seem to focus on "disaster relief" (for example, p. 2, line 93; or the example of the earthquake in Wenchuan). It is important to clarify which one of the two perspectives you consider under the concept of PCSR, specially in relation with your hypothesis 2a. There, when you describe the behavior of individualistic organizations, it makes sense if by PCSR you understand single reactions to particular disasters, but it is less clear if you understand by PCSR "contribution to sustainable development", which involves a more long-term perspective, that does not fit with your description of how individualistic companies behave. On the other hand, if by PCSR you mean "involvement in disasters relief", then you need to justify better this view, because you are taking one aspect for the whole. 

3. Literature review. A few comments in this section:

- You start this section by saying that "three perspectives have been used" (line 74). Which are these three perspectives. I am a little bit lost with all the classifications you provide. 

- You continue by saying that "previous research noted that corporate identity consists of normative and utilitarian components" and you spend the rest of the paragraph elaborating on this distinction. In the next page, you present the Organizational Identity Orientation theory and the three types of identity. It is confusing the use of the term "identity" in both cases. Is there any relation between them? If not, wouldn't be better to use different "terms" and not "identity" in both cases?

- The last paragraph of this section (p.3, lines 95-109) is a review of another framework: instrumental, normative, moral. I was wondering why were you mentioning this framework since you have not make any use of it during the rest of the paper. At the very end of your paper (page 13, lines 500ff.), you mention again this framework to explain the motivation of companies for CSR. I suggest that you explain better in the literature section why are you introducing this framework, and how does it fit with your whole framework. Maybe you can move the explanation in page 13 here, since these two paragraphs are more a kind of explanation of the framework than a theoretical implication of your paper.

- if you are talking about theories that explain the motivation of companies for CSR, I don't understand why you mention three theories in the literature review (instrumental, normative and moral), and you leave the fourth one (virtue ethics theory) for the next section. Wouldn't this fourth theory be part of the literature review.

- Finally, as an overall comment, I am not sure you can say that this section is a literature review. It is more the theoretical framework explained, together with the first part of the next section.

4. Well-intentioned vs. higly-motivated companies.  You introduce this distinction in a very soft manner in page 3 (lines 131-135). But, at the end of your paper, this distinction seems to grow in importance. Even you seem to use this distinction to reformulate your research question (page 11, lines 441-442). If this is so, you need to review how do you formulate your research question, and you need to provide a long explanation of this concept of "genuineness" (line 131). 

The three types of identity. I have several doubts on how do you describe the three types of OIO in your elaboration of the hypothesis:

-      regarding individualistic identity, as I have already mentioned above, the "instrumental" character of this type does not fit with a broad description of Public CSR as pursuing the common good.

-      Regarding relational identity, is this dyadic relationship so strong and focused? Only relation in a vis-à-vis manner with another stakeholder. What about if this other stakeholder is the whole society?

-      Regarding collectivistic-identity. If this type is described as a multi-stakeholder relation based on a common good, it does not fit at all with your reference that this type implies a weak tie with stakeholders. Is the pursuit of the common good a weak tie? This needs to be better justified.

5. Method.  I must admit that my knowledge of the statistical apparatus is limited, and I cannot give you too much feedback here. I hope that other reviewers may. However, I would like to see a more detailed description of the questionnaire.

6. Can a virtuous company be individualistic? When you were describing the four models of your empirical research, a doubt came to my mind. If we describe a "virtuousness organization" as a set of five virtues, how can such a company be individualistic? Or even relational?  You mentioned in a line (page 4, line 172) that "organizations are not totally virtuous or non-virtuous", which I think explains why a so-called (not completely) virtuous organization can be individualistic or relational, but I would suggest that you elaborate a little bit this possible objection.

7. On the "managerial implications" subsection, you devote a long paragraph describing eight parameters that –to my understanding- have nothing to do with your research (at least they are not implied by your findings). I suggest that you review this section and find more managerial implications (also take into account next point)

8. Instrumental vs moral/individualistic vs collectivistic? According to your findings, there is a positive effect of both the individualistic and the collectivistic identity. In page 12 (lines 476ff.) you seem to identify individualistic identity with instrumental motivation (I guess that you identify also collectivistic identity with moral motivation).  My concern here is that, although in effective terms both can have a positive impact on PCSR, it is very different that you engage in PCSR because of an instrumental motivation or because of a moral one, and your study does not distinguish it. Maybe this is a limitation or a question to investigate in further research.

9. Is Vanke really a virtuousness and relational company? It is good that you try to bring your example at the end of your paper again. But I do not think it is fair that you describe Vanke as a virtuousness and relational company just because it fits with your framework and without providing more empirical support. I can find other explanations of the public reaction against Vanke. for example, they had the bad fortune that another smaller company offered a bigger quantity for disaster relief. What about if no other company has appeared? Maybe the public had been happy with the 20 Million donation… 

10. Are these really the limitations of your paper? I know that a paper has to finish with a section on limitations and further research. But if I have to take seriously your comments, and you really describe your paper as "small size sample", "simplistic approach in aggregating individual data" and "biased and self-reporting", it seems that you are asking me to reject your paper, and I don't want to do it. So, please, make an effort of mentioning relevant and real limitations of your paper, and do not use common places. 

I hope these comments make you thinking again on your paper and make it stronger. It really deserves it. 

Author Response

1. Sufficient and necessary conditions? In several moments of their text they mention that "organizational virtuousness" (OV) is a sufficient but not necessary condition for Public CSR (PCSR).  Based on the Research Model (Figure 1) they want to investigate the relation between OV and PCSR, and they include as "moderator factors" the three types of Organizational Identity Orientation (OIO). To my understanding, they want to test that OV is not enough in order to explain why a company engages or succeeds in PCSR, but it is also necessary to include other variables such as OIO. Therefore, what the models says is that OV is necessary but not sufficient, right? If I am right, then you need to review several sentences along the text (for example, abstract, or p. 12, lines 451-454).

Response 1: We have reorganized the logical structure of the paper.

2. Does Public CSR mean engaging in disaster relief?. There is a confusing use of the concept of Public CSR along the text. In certain moments, the authors seem to refer to PCSR as how the companies contribute to sustainable development (for example, p2, line 83; or p.5, line 196: "Public CSR is just the so-called "societal benefit" and "common good" mentioned above"), but in other moments they seem to focus on "disaster relief" (for example, p. 2, line 93; or the example of the earthquake in Wenchuan). It is important to clarify which one of the two perspectives you consider under the concept of PCSR, specially in relation with your hypothesis 2a. There, when you describe the behavior of individualistic organizations, it makes sense if by PCSR you understand single reactions to particular disasters, but it is less clear if you understand by PCSR "contribution to sustainable development", which involves a more long-term perspective, that does not fit with your description of how individualistic companies behave. On the other hand, if by PCSR you mean "involvement in disasters relief", then you need to justify better this view, because you are taking one aspect for the whole. 

Response  2: This paper focuses on public CSR. Disaster relief is only an example of public CSR. This paper tries to illustrate the determinants of public CSR through this example. We have reorganized the logical structure of the paper. For more details, please refer to the relevant parts of the paper.

3. Literature review. A few comments in this section:

- You start this section by saying that "three perspectives have been used" (line 74). Which are these three perspectives. I am a little bit lost with all the classifications you provide. 

Response 3: We have reorganized the logical structure of the paper.

- You continue by saying that "previous research noted that corporate identity consists of normative and utilitarian components" and you spend the rest of the paragraph elaborating on this distinction. In the next page, you present the Organizational Identity Orientation theory and the three types of identity. It is confusing the use of the term "identity" in both cases. Is there any relation between them? If not, wouldn't be better to use different "terms" and not "identity" in both cases?

Response: The concept of identity is different from identity orientation. Organizational identity typically refers to those elements that members deem central, distinctive, and enduring about an organization (Albert and Whetten, 1985; Pratt and Foreman, 2000), while organizational identity orientation refers to the nature of assumed relations between an organization and its contractual or immediate stakeholders—are relations independent, dyadically interdependent, or derived from a common group membership? Identity orientation construct may provide insight into the link between organizational identity and organizations’ relations (Brickson, 2005).

- The last paragraph of this section (p.3, lines 95-109) is a review of another framework: instrumental, normative, moral. I was wondering why were you mentioning this framework since you have not make any use of it during the rest of the paper. At the very end of your paper (page 13, lines 500ff.), you mention again this framework to explain the motivation of companies for CSR. I suggest that you explain better in the literature section why are you introducing this framework, and how does it fit with your whole framework. Maybe you can move the explanation in page 13 here, since these two paragraphs are more a kind of explanation of the framework than a theoretical implication of your paper.

Response : We have reorganized the logical structure of the paper.

- if you are talking about theories that explain the motivation of companies for CSR, I don't understand why you mention three theories in the literature review (instrumental, normative and moral), and you leave the fourth one (virtue ethics theory) for the next section. Wouldn't this fourth theory be part of the literature review.

Response : We have reorganized the logical structure of the paper. We start our inquiry with the following question: why would a corporation be willing to take public CSR? In reviewing the CSR literature, instrumental, normative and moral perspectives have been used to explain why for-profit organizations will make voluntary contributions to serve public purposes. While, in our opinion, activities such as CSR behavior can be seen as resulting not only from external demands, but also from the corporations’ internal personality, namely the activities are the external expression of corporations’ internal personality trait. In this study, we intend to reveal the motivation of corporations to take public CSR from the perspective of organizational traits with evidence fromChina.

- Finally, as an overall comment, I am not sure you can say that this section is a literature review. It is more the theoretical framework explained, together with the first part of the next section.

Response : We have reorganized the logical structure of the paper.

4. Well-intentioned vs. higly-motivated companies.  You introduce this distinction in a very soft manner in page 3 (lines 131-135). But, at the end of your paper, this distinction seems to grow in importance. Even you seem to use this distinction to reformulate your research question (page 11, lines 441-442). If this is so, you need to review how do you formulate your research question, and you need to provide a long explanation of this concept of "genuineness" (line 131). 

Response 4: We have reorganized the logical structure of the paper.

The three types of identity. I have several doubts on how do you describe the three types of OIO in your elaboration of the hypothesis:

-      regarding individualistic identity, as I have already mentioned above, the "instrumental" character of this type does not fit with a broad description of Public CSR as pursuing the common good.

-      Regarding relational identity, is this dyadic relationship so strong and focused? Only relation in a vis-à-vis manner with another stakeholder. What about if this other stakeholder is the whole society?

-      Regarding collectivistic-identity. If this type is described as a multi-stakeholder relation based on a common good, it does not fit at all with your reference that this type implies a weak tie with stakeholders. Is the pursuit of the common good a weak tie? This needs to be better justified.

Response : Generally speaking, stakeholders of a firm can be divided into contractual stakeholders and community stakeholders. Contractual stakeholders have direct or financial relationship with the firm.  Community stakeholders, such as the local community or society, or the public have indirect relationship with the firm. The definition of OIO is related to contractual or direct stakeholders. If a company views itself primarily as being distinctive from its competitive counterparts, and keeps independent relation with its contractual stakeholders, it is individualistic-identity oriented; if a company views itself primarily as a good partner of those with whom it interacts, and keeps dyadically interdependent relationship with its contractual stakeholders, it is relational-identity oriented; yet if a company views itself primarily a good member to a larger community and works to improve the welfare of the community it values and/or belongs to, it is collectivistic-identity oriented (Brickson, 2005).

5. Method.  I must admit that my knowledge of the statistical apparatus is limited, and I cannot give you too much feedback here. I hope that other reviewers may. However, I would like to see a more detailed description of the questionnaire.

Response 5: Amendments have been made as required. For more details, please refer to the relevant parts of the paper.

6. Can a virtuous company be individualistic? When you were describing the four models of your empirical research, a doubt came to my mind. If we describe a "virtuousness organization" as a set of five virtues, how can such a company be individualistic? Or even relational?  You mentioned in a line (page 4, line 172) that "organizations are not totally virtuous or non-virtuous", which I think explains why a so-called (not completely) virtuous organization can be individualistic or relational, but I would suggest that you elaborate a little bit this possible objection.

Response 6: According to (Brickson, 2005), organizational identity orientation refers to the nature of assumed relations between an organization and its contractual or immediate stakeholders—are relations independent, dyadically interdependent, or derived from a common group membership? A virtuous corporation has many virtues. These virtues interact with each other, which together make the corporation has a virtuous character and show an ethos of virtuousness. Organizational virtuousness is a different concept from OIO, it is reasonable for virtuous companies to have different identity orientations.

7. On the "managerial implications" subsection, you devote a long paragraph describing eight parameters that –to my understanding- have nothing to do with your research (at least they are not implied by your findings). I suggest that you review this section and find more managerial implications (also take into account next point)

Response 7: The results of this study show that, in general, organizational virtuousness has a positive impact on public CSR behavior. So, to promote a company’s motivation in taking public CSR, it is critical to cultivate organizational virtuousness. About how to cultivate organizational virtuousness, we give a suggestion that designing an internal governance system in the corporation.

8. Instrumental vs moral/individualistic vs collectivistic? According to your findings, there is a positive effect of both the individualistic and the collectivistic identity. In page 12 (lines 476ff.) you seem to identify individualistic identity with instrumental motivation (I guess that you identify also collectivistic identity with moral motivation).  My concern here is that, although in effective terms both can have a positive impact on PCSR, it is very different that you engage in PCSR because of an instrumental motivation or because of a moral one, and your study does not distinguish it. Maybe this is a limitation or a question to investigate in further research.

Response 8: Collectivistic-identity oriented organizations see themselves as a member of a larger group such as the society or community. This type of organizations will forge external and internal contractual stakeholder relationships based on a common goal. Although both collectivistic and individualistic companies view the relationship with immediate stakeholders as a means to an end, individualistic organizations use the relationship to meet their self-defined objectives, while collectivistic organizations use the relationship to meet common goals (Brickson, 2007).

9. Is Vanke really a virtuousness and relational company? It is good that you try to bring your example at the end of your paper again. But I do not think it is fair that you describe Vanke as a virtuousness and relational company just because it fits with your framework and without providing more empirical support. I can find other explanations of the public reaction against Vanke. for example, they had the bad fortune that another smaller company offered a bigger quantity for disaster relief. What about if no other company has appeared? Maybe the public had been happy with the 20 Million donation… 

Response 9: Our research draws a conclusion that virtuous corporation is not necessarily more willing to take public CSR, to some degree it depends on the type of identity orientation possessed by the virtuous corporation. This conclusion can be illustrated by an example of so-called “donation-gate” crisis. We have reorganized the logical structure of the paper.

10. Are these really the limitations of your paper? I know that a paper has to finish with a section on limitations and further research. But if I have to take seriously your comments, and you really describe your paper as "small size sample", "simplistic approach in aggregating individual data" and "biased and self-reporting", it seems that you are asking me to reject your paper, and I don't want to do it. So, please, make an effort of mentioning relevant and real limitations of your paper, and do not use common places. 

Response 10: Amendments have been made as required. We have added future research directions in this paper. For more details, please refer to the relevant parts of the paper.

I hope these comments make you thinking again on your paper and make it stronger. It really deserves it. 

Thank you.

Round 2

Reviewer 1 Report

Abstract: Instead of "Corporation social responsibility" please use "Corporate Social Responsibility"

Introduction: Authors explaination regarding the distinction into 'relational' and 'public CSR' are for me not convincing. In lines 36-38 in author's words: 

" The relational CSR is more or less transactional, instrumental, and compulsory, whereas the public CSR is in the ethical, 37 philanthropic, and discretional domain (Albareda, Lozano, & Ysa, 2007). "

Whereas  Albareda, Lozano and Ysa (2007, p. 395) define relational CSR as follow:

"Relational CSR: CSR public policies designed to improve collaboration between governments, businesses and civil society stakeholders."

The purpose of the article of Albareda, Lozano and Ysa (2007,) refers to different CSR public policies adopted by European governments in order to promote responsible and sustainable business practices.

I can hardly agree that this citation supports the understanding of CSR in terms provided in reviewed article. If so, please provide the specific page number, where I should look for the explaination. 

In my opinion authors give new name for old phenomena. I would suggest to explain the framework on existing theories,  (eg. Carroll 1979 - philantropic dimension, Porter and Kramer - strategic Philantropy) and use existing tems and models (e.g. the secondary groups of stakeholders, ) in order to explain the research instade of creating "new" distinction without proper and clear theoretical justification. In my opinion authors failed in providing a clear justification for the relational and public CSR. I suggest to work on the justification,  rewrite using existing theories for explaining public CSR or leave the relational CSR and concentrate on the public CSR (clearly defined)

In order to explain deeper my negative opinion on the way the 'relational CSR' was defined by authors I would like to point at the fact, that this understaning contradicts the core characteristics of relational Organizational Identity (Brickson 2005, 2007). This lack on consistent communication leads to misinterpretation and is misleading for the reader. 

 Line 69-70" corporations’ internal personality, namely the activities 69 are the external expression of corporations’ internal personality trait. 

I suggest to resign from the world 'personality' and use rather the "identity' concept.

Theoretical Framework. 

In this part authors explain the relationships between measured concepts, yet they don't define clearley these elements of the model:

- public CSR, OIO and Organizational virtousness

First the undersdanding appears while these are operationalized. Anyway I would expect to have a clear definitions in the theory section. 

The empirical part of the paper is much better. It is much more clear, however the first part of conclusions is no really a conclusion, but an example of how it may be perceived in the practice. 

General comment

In general the improvements in the paper help to make the paper more clear, anyway I think authors too often split their narrative, jump from theory to theory and unfortunatelly try to tell too many things at once, so that the theory part makes an impression of inconsistent and for the reader it is not easy to keep truck. As I'm interested in this topic I think I understand what authors are trying to say. And it would be interesting in authors one more time read the theory part and try to take out/change aspects, that are controverse and force reade to 'belive' the authors. Insteade authors could keep/ develop consistant explainations that are important for the paper.   In my opinion it may be possible when authors stick to the main aim of the paper (what they are trying to tell in the paper) and concentrate on saying "why is it so". I see potential in the paper, and I belive it can be improved so that both" autors and readers will benefit. 

Reviewer 2 Report

The paper has been significantly improved and the underlined limitations have been successfully addressed.

Reviewer 3 Report

Dear authors,

Thank you for trying to take into account my comments in your new version of the paper. In my opinion, the text has improved significantly, although it requires a few more things to review. Minor revisions, in any case.

1. let me start with the title. You have introduced the concept of "organizational traits". If you have decided to make more explicit this issue, then you need to explain better what does it mean. Maybe in the introduction or in the theoretical framework, you should devote more space to explain what the concept of "trait approach" means. 

2. Abstract. The second part of the abstract –all the text you have added from the original paper- has too many concepts. For someone not familiar with these concepts, this text could be difficult to understand. Try to make it more digestible. 

3. Sometimes moving paragraphs to a different location is a good solution; but not always. And, in any case, it is not enough with moving paragraphs: the final text needs to have some consistency. You have rewritten the introduction moving some paragraphs from other places, and you have also removed the case of Vanke. This makes the Introduction more theoretical, and a little bit more consistent, but I would recommend that you try to help the reader to enter your text more smoothly. The first lines (with the reference to corporate identity, the two components, and the different foci) are too direct. 

3. At the end of the introduction, the mention to "evidence from China" should be a little bit more explained. 

4. There is still some confusion on the research question. In the introduction (page 2, line 71) you say that you intend to "reveal the motivation of corporations to take public CSR from the perspective of organizational traits". Immediately afterward (page 3, line 79), you refer to the issue of "what kind of corporations with what traits are more willing to take public CSR". Is it the same research question just rephrased? If this is the case, it is different to study the "motivation" or study the "traits". Which one is your question? Otherwise, is the second sentence a subset of the main research question? If this is the case, you should clarify the relationship between these two questions.

5. Thanks for detailing better the content of the survey.

6. The discussion section. First of all, regarding the whole structure of this section, you have decided to include everything there: conclusions from your survey, theoretical and practical implications, limitations and further directions. Please, consider whether it is better to split all this content into different sections. 

7. In the Discussion subsection, you introduce the Vanke case. I am sorry to say that you are judging the intentions of Vanke's managers in such a way that it is not fair. How do you know that "the donation is sincere", or that "its willingness to contribute was weaker" ort that the "Vanke group takes the interests of its contractual stakeholders more into account", or that the "Vanke Group is a highly relational-identity oriented company"? All these strong affirmations have not been proved at all. I am afraid that if you delete this example, the subsection will be too shorter. So you need to be creative on how to make a "discussion" section really relevant.

8. I am not going to repeat my doubts regarding Moore's 8 parameters, but at least try to make more explicit why do you bring this example here. You were more explicit in your answer to my suggestions.

9. The first paragraph of the "theoretical implications" is a very short and direct reference to the trait approach. I would suggest that you develop a little bit longer this point. Remember that this is now your main focus (according to your new title). The second paragraph of this section is a good explanation of the trait. I do not understand why do you leave it for the very end of your paper, whereas this explanation of "trait approach" should be given from the very beginning. But again, if you move this paragraph, this section will be only a single and short paragraph; you should rethink how you make this section more robust.

And that's all. I hope these comments help you to keep improving your paper. They are a few, but they require a minor effort, I guess.